# The epidemiology and disease burden of congenital TORCH infections among hospitalized children in China: A national cross-sectional study

**Linlin Zhang**[1,2], **Xinyu Wang**[3], **Mengjia Liu**[1,2], **Guoshuang Feng**[3], **Yueping Zeng**[4], **Ran Wang**[1,2]*, **Zhengde Xie**[1,2]*

**1** Beijing Key Laboratory of Pediatric Respiratory Infectious Diseases, Key Laboratory of Major Diseases in Children, Ministry of Education, National Clinical Research Center for Respiratory Diseases, Laboratory of Infection and Virology, Beijing Pediatric Research Institute, Beijing Children's Hospital, Capital Medical University, National Center for Children's Health, Beijing, China, **2** Research Unit of Critical Infection in Children, 2019RU016, Chinese Academy of Medical Sciences, Beijing, China, **3** Big Data Center, Beijing Children's Hospital, Capital Medical University, National Center for Children's Health, Beijing, China, **4** Medical Record Management Office, Beijing Children's Hospital, Capital Medical University, National Center for Children's Health, Beijing, China

* randall@mail.ccmu.edu.cn (RW); xiezhengde@bch.com.cn (ZX)

**Data Availability Statement:** All relevant data are within the manuscript and its Supporting information files.

## Abstract

### Background

Congenital TORCH (*Toxoplasma gondii (T. gondii)*, rubella virus (RV), cytomegalovirus (CMV), and herpes simplex virus (HSV)) infections are associated with a variety of adverse prenatal and neonatal events, including miscarriage, malformations and developmental abnormalities, and they remain an issue that cannot be neglected in China. However, the current research focuses more on the general screening of TORCH in women of childbearing age, and the medical information of children hospitalized due to congenital and perinatal TORCH infections has not been described in detail. This study summarized and analyzed the epidemiological characteristics, clinical manifestations, length of stay (LOS), and the disease burden of hospitalized children diagnosed with congenital TORCH infections in 27 children's hospitals in China.

### Methodology

Based on the face sheet of discharge medical records (FSMRs) of hospitalized children in 27 tertiary children's hospitals collected in the Futang Research Center of Pediatric Development and aggregated into FUTang Update medical REcords (FUTURE), we summarized and analyzed the epidemiological characteristics, clinical manifestations, LOS, the disease burden (in US dollars, USD) and potential risk factors for hospitalized children diagnosed with congenital toxoplasmosis, congenital rubella syndrome, congenital cytomegalovirus infection, and congenital HSV in 27 children's hospitals in China from 2015 to 2020.

**Funding:** National Natural Science Foundation of China (No.82002130) and Beijing Natural Science Foundation (No.7222059) were awarded to R.W. The CAMS Innovation Fund for Medical Sciences (No.2019-I2M-5-026) was awarded to ZD. X. The funders had no role in study design, data collection and analysis, decision to publish, or preparation of the manuscript.

**Competing interests:** The authors have declared that no competing interests exist.

## Results

One hundred seventy-three patients aged 0–<1 year were hospitalized for congenital TORCH infections. Among infections with TORCH, hospitalization with congenital toxoplasmosis was the least common, with only five cases were reported (2.89%), while the LOS was the highest. The proportion of patients with congenital rubella syndrome (CRS) was 15.61%, and 86% of children hospitalized with CRS had cardiovascular malformations, and the economic burden was the highest. Congenital CMV infection cases accounted for the largest proportion (76.30%). Overall, 5.20% of patients were infected with HSV, and the expense of hospitalization for congenital HSV infection was relatively low.

## Conclusion

In the present study, the hospitalization proportion due to congenital TORCH infection was extremely low (17.56 per 100,000 neonates), indicating that China's congenital TORCH infection prevention and control policies remain effective. The lowest proportion of patients was hospitalized with congenital toxoplasmosis, while the LOS was the longest. The economic burden of CRS was heavy, and infants are recommended be vaccinated against RV in a timely manner. Congenital CMV infections accounted for the largest proportion of patients, suggesting that the disease burden of congenital CMV infection cannot be ignored, and the prevention of congenital CMV infection during pregnancy is still an important issue that needs to pay attention. The expense of hospitalization for congenital HSV infection was relatively low, while the disease burden increases significantly when patients develop complications. These data illustrate the importance of improving screening for congenital TORCH infections in the early diagnosis and treatment of neonatal patients.

## Author summary

Congenital TORCH (*T. gondii*, RV, CMV, and HSV) infections are associated with a variety of adverse prenatal and neonatal events, including miscarriage, malformations, and developmental abnormalities. Congenital TORCH infections remain an issue that cannot be neglected in China. However, the current research focuses more on the general screening of TORCH in women of childbearing age, and the medical information from children hospitalized due to congenital TORCH infections has not been described in detail. This study summarized and analyzed the epidemiological characteristics, clinical manifestations, length of stay (LOS), and disease burden of hospitalized children diagnosed with congenital toxoplasmosis, congenital rubella syndrome (CRS), congenital cytomegalovirus infection, and congenital herpes simplex virus infection in 27 children's hospitals in China from December, 2015 to December, 2020 based on the FUTang Update medical Records (FUTURE). A total of 173 patients were hospitalized for congenital TORCH infections. The hospitalization rate due to congenital TORCH infection was extremely low (17.56 per 100,000 neonates), showing that China's congenital TORCH infection prevention and control policies remain effective. Hospitalization due to congenital toxoplasmosis was the least common, with only five cases reported (2.89%). The economic burden of CRS was heavy, and infants are recommended be vaccinated against RV in a timely manner. Congenital CMV infections accounted for the largest proportion of cases, suggesting that the disease burden of congenital CMV infection cannot be ignored, and the

prevention of congenital CMV infection during pregnancy is still an important issue that needs to pay attention. The hospitalization expense of congenital HSV infection was relatively low, while the disease burden increases significantly when patients develop complications. These data illustrate the importance of improving screening for congenital TORCH infections in the early diagnosis and treatment of neonatal patients. A complete understanding of the epidemiology and disease burden of congenital TORCH infections is essential for reducing the incidence of adverse pregnancy outcomes.

## Introduction

The acronym 'TORCH' was introduced by Nahmias in 1971 to underline a group of pathogens that cause congenital and perinatal infections: *Toxoplasma gondii* (*T. gondii*), rubella virus (RV), cytomegalovirus (CMV), and herpes simplex virus (HSV) [1]. Congenital TORCH infections are significant cause of prenatal and neonatal abnormalities and deaths. Apart from miscarriage, stillbirths, and neonatal deaths, congenital TORCH infections account for 2% to 3% of all congenital anomalies [2].

*T. gondii* is the only zoonotic protozoan parasite among TORCH pathogens. Toxoplasma infection usually occurs through contact with feces of infected cats or consumption of food or water contaminated with the parasite. *T. gondii* infection has a wide distribution, and the estimated global incidence of congenital toxoplasmosis is approximately 190,100 cases annually, with an incidence rate of approximately 1.5 cases per 1,000 live births [3]. Infection with *T. gondii* during pregnancy results in severe fetal damage including hydrocephalus, intracerebral calcifications, mental retardation, chorioretinitis, and death [4]. Studies have reported that 29.8% of congenital toxoplasmosis cases might result in eye damage [5], and the disease burden caused by this disease must not be overlooked.

Primary RV infection in the early stages of pregnancy may lead to serious birth defects known as congenital rubella syndrome (CRS). Congenital heart defects (such as patent ductus arteriosus and branch pulmonary artery hypoplasia/ stenosis), cataracts, and sensorineural hearing loss are the classic triad. The WHO estimates that approximately 100,000 congenital rubella syndrome cases occur per year [6, 7]. RV is a teratogenic virus, and the fetus teratogenic rate reaches 10–30%, which places a heavy economic and social burden on the family and society [8]. Since the inclusion of rubella in the China Expanded Programme on Immunization (EPI) system in 2008, the incidence of rubella has decreased from 91.00 per million in 2008 to 2.83 per million in 2018 [9]. However, there were some studies reported that the seropositivity of anti-rubella IgG antibodies in 8-month-old infants was only 3.33–4.6% in several regions of China. Thus, the protection of maternally transferred antibodies is still insufficient, and studies of the epidemiology and disease burden of CRS in neonates are necessary.

CMV is currently recognized as the most common virus causing intrauterine infection, with a reported global prevalence of congenital CMV infection of approximately 0.2% to 2% [10]. China is the high-incidence area for CMV infection. A study found that the positive rate of CMV-IgM in newborns and infants in Qingdao, China was 0.67%, and the positive rate of IgG was 96.28% [11]. Infants with symptomatic congenital CMV infection are at significantly increased risk of developing adverse long-term outcomes, the most common manifestations are jaundice, congenital malformations, and liver damage, and other clinical manifestations include premature birth, central nervous system involvement, neuroimaging anomalies chorioretinitis, and sensorineural hearing loss [12]. Congenital CMV infection seriously endangers the health and safety of children.

Vertical transmission is one of the main transmission routes of congenital HSV infection. An estimated 1/3,200 live births are infected with HSV [13]. Eighty percent of infants with the disseminated disease die without treatment, and those who survive tend to have severe brain damage, which places a heavy burden on the public health system and society [14]. Ambroggio *et al.* estimated that hospitalization costs up to 37,431 US dollars (USD) per patient with neonatal HSV infection (interquartile range: 14,667–74,559 USD) [15]. However, due to a lack of reported data on congenital HSV infection, its disease burden is unclear in China.

The introduction of routine maternal screening for TORCH in China has substantially reduced the overall burden of infection during pregnancy and substantially reduced adverse neonatal outcomes [16]. However, congenital TORCH infections still cause substantial morbidity. Some research reported that the seroprevalence of TORCH infections in Chinese pregnant women was 6.06%, and the positive rates of TORCH-IgM and IgG in newborns were 0.78% and 99.8%, respectively [11, 17]. Congenital TORCH infection remains an issue that must not be neglected in China. However, the current research focuses more on the general serologic screening of TORCH in women of childbearing age [7, 16–18]. The epidemiology and economic burden of hospitalized children due to congenital TORCH infections have not been described in detail, and few therapeutic options are available to manage these infections. The face sheet of discharge medical records (FSMRs) in the electronic medical system has generated a large amount of medical data for all patients during hospitalization, which has the value of evidence-based medicine. Based on the FUTang Update medical Records (FUTURE) [19], a national pediatric patient database in China, this study summarized and analyzed the epidemiological characteristics, clinical manifestations, length of stay (LOS), disease burden (in USD) and potential risk factors for hospitalized children diagnosed with congenital toxoplasmosis, CRS, congenital CMV infection, and congenital HSV infection in 27 children's hospitals in China from 2015 to 2020. Understanding the epidemiology and disease burden of congenital TORCH infections among hospitalized children is important to inform the optimization of care pathways and utility newborn screening.

## Methods

### Ethics statement

This study was approved by the Ethics Committee of Beijing Children's Hospital, Capital Medical University (Approval Number: [2022]-E-008-R). The requirement for informed consent of patients was waived as it only involved retrospective aggregated data analysis of medical records. Our data were fully deidentified and anonymized to protect privacy.

### Data source

The Futang Research Center of Pediatric Development (FRCPD) is the first nonprofit social service organization to engage in pediatric development research under the supervision and management of the Ministry of Civil Affairs of China. The FRCPD has built a health service network system with 47 provincial children's medical institutions as the core to strengthen the data-sharing connection among the member hospitals of the Futang Children's Medical Development Research Center and improve the efficiency of data utilization.

FUTURE covers the face sheet of discharge medical records (FSMRs) of hospitalized children in 27 tertiary children's hospitals in FRCPD. The FRCPD started to collect FSMRs data in December 2015, and the staff at FRCPD were responsible for checking and validating the uploaded data to control its quality and integrity.

## Study design

The data analyzed in this study were retrospectively extracted from the FUTUR database for the variables of congenital toxoplasmosis, CRS, congenital CMV infection, and congenital HSV infection from 2015 to 2020. We also analyzed sociodemographic and geographic variables, admission and discharge information, primary and secondary diagnoses and hospitalization expenses of the patients. We reported the epidemiology of the cohort by calculating the total to points for the type of virus with which patients were infected and stratified them by gender, age groups, geographic regions, ethnicity, and clinical manifestation. Among them, hospitalized patients were divided into two groups according to their age (day, d, or year, y): 0–28 d (neonate) and 29 d–<1 y (infant). The 27 children's hospitals were grouped into seven geographic regions: Northeast, North, East, South, Central, Northwest, and Southwest China [20, 21].

## The eligibility of the participants and admission records

The inclusion and exclusion criteria for the participants were as follows:

1. All included patients who were admitted with a primary diagnosis of any congenital toxoplasmosis, CRS, congenital CMV infection, or congenital HSV infection between December 1, 2015, and December 31, 2020. The 10[th] Revision of International Statistical Classification of Diseases and Related Health Problems (ICD-10) codes was used as the selection criteria for the primary screening and classification of disease.

2. All included patients had a clear pathogenic and clinical diagnosis (according to the diagnostic criteria promulgated by the Health Commission of the People's Republic of China (formerly the Ministry of Health), (http://www.nhc.gov.cn/fzs/s7852d/201512/2155fc4055ce4df687536233e1639729.shtml, http://www.nhc.gov.cn/wjw/s9492/201409/dfb9e7068025408c820e227342f7ac8b.shtml), *Practical Neonatology* and *Expert consensus on standardized TORCH laboratory detection and clinical application* (http://rs.yiigle.com/CN114452202005/1199015.htm).

3. Patients with incomplete information for multiple vital data, such as patient gender, age, diagnosis, or burden of disease, were excluded. Patients diagnosed with other viral infections were excluded.

4. The patients were aged ≤18 years on the index admission.

## Statistical analysis

Categorical variables are described as percentages. Nonnormally distributed continuous variables are described as medians and interquartile ranges (IQRs). Differences between two groups were compared with either the chi-squared test (unordered categorical variables) or the Mann–Whitney–Wilcoxon test (ordered categorical or nonnormal-distributional continuous variables), as appropriate. The nonparametric Kruskal–Wallis test was used for comparisons among multiple groups. All statistics were analyzed using SPSS software version 22.0 (SPSS Inc., USA). Differences with $p$ values<0.05 were considered statistically significant.

## Result

### Overall

According to the ICD-10 codes, we searched for hospitalizations with the first diagnosis of congenital toxoplasmosis, CRS, congenital CMV infection, and congenital HSV infection

**Table 1. The general sociodemographic characteristics of pediatric patients with TORCH infections during hospitalization from December 2015 to December 2020.**

| Categories | TORCH infections (n, %) | | | |
|---|---|---|---|---|
| | *T. gondii* | RV | CMV | HSV |
| **Overall (n, %)** | 5 (2.89) | 27 (15.61) | 132 (76.30) | 9 (5.20) |
| **Sex (n, %)** | | | | |
| **Male** | 3 (60.00) | 13 (48.15) | 74 (56.06) | 5 (55.56) |
| **Female** | 2 (40.00) | 14 (51.85) | 58 (43.94) | 4 (44.44) |
| **Age (n, %)** | | | | |
| 0–28 d | 4 (80.00) | 26 (96.30) | 64 (48.48) | 6 (66.67) |
| 29 d–<1 y | 1 (20.00) | 1 (3.70) | 68 (51.52) | 3 (33.33) |
| **Residence (n, %)** | | | | |
| Urban | 2 (40.00) | 10 (37.04) | 57 (43.18) | 3 (33.33) |
| Rural | 3 (60.00) | 17 (62.96) | 75 (56.82) | 6 (66.67) |
| **LOS [d, median (IQR)]** | 19 (4.5–23) | 9 (5–15) | 10 (6–20.75) | 6 (2.5–18.5) |
| **Expense [USD, median (IQR)]** | 2,162.33 (637.51–3,523.25) | 2,585.45 (1,524.88–4,010.98) | 1,379.00 (823.64–3,268.45) | 935.03 (505.80–3,982.70) |

*T. gondii*: *Toxoplasma gondii*

RV: Rubella Virus

CMV: Cytomegalovirus

HSV: Herpes Simplex Virus

LOS: length of stay

IQR: inter quartile range

USD: USA dollar

(congenital TORCH infections) under the FUTURE database. As showed in Table 1, 173 patients were hospitalized for congenital TORCH infections. Among infections with any pathogen causing TORCH, congenital CMV infection cases accounted for the largest proportion (132 cases, 76.30%), followed by CRS cases (27 cases, 15.61%), and nine cases of congenital HSV infection (5.20%). Patients hospitalized with congenital toxoplasmosis represented the lowest proportion, and only five cases were reported (2.89%).

We analyzed the proportions of congenital TORCH infection-related hospitalization to total hospitalizations within different age groups of patients in the corresponding groups. As shown in Fig 1A, the proportion of hospitalizations for congenital TORCH infections was 17.56/100,000 hospitalized newborn (100/569,680) and 12.47/100,000 hospitalized infant (73/585,244). Congenital toxoplasmosis cases accounted for 0.70/100,000 hospitalized newborn (4/569,680) and 0.17/100,000 hospitalized infant (1/585,244), CRS cases accounted for 4.56/100,000 hospitalized newborn (26/569,680) and 0.17/100,000 hospitalized infant (1/585,244), congenital CMV infection cases accounted for 11.3/100,000 hospitalized newborn (64/569,680) and 11.62/100,000 hospitalized infant (68/585,244), and congenital HSV infection cases accounted for 1.05/100,000 hospitalized newborn (6/569,680) and 0.51/100,000 hospitalized infant (3/585,244), respectively. These results showed that the proportion of hospitalizations due to congenital TORCH infection is low, and the age at which patients were hospitalized was mainly in the neonatal period, except for congenital CMV infection.

Next, we analyzed the proportion of congenital TORCH infections hospitalizations to total hospitalization included in this study by region in different age groups. In the cases of neonates, patients from the Central China had the highest proportion of hospitalization (0.36‰, 24/67,524) while no neonates hospitalized with congenital TORCH infection in the Northeast China (Fig 1B). The hospitalization proportions of neonate patients from North China, East

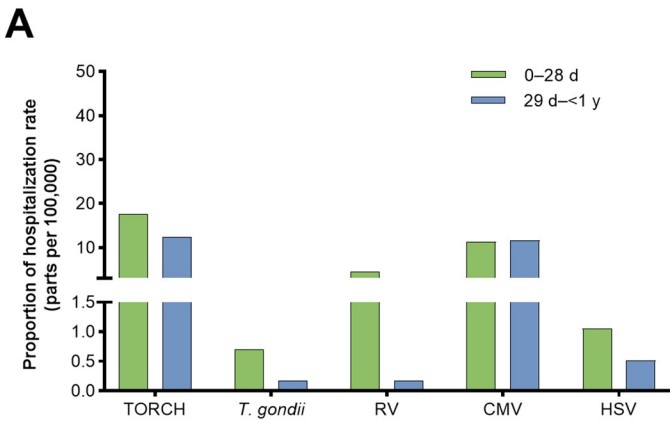

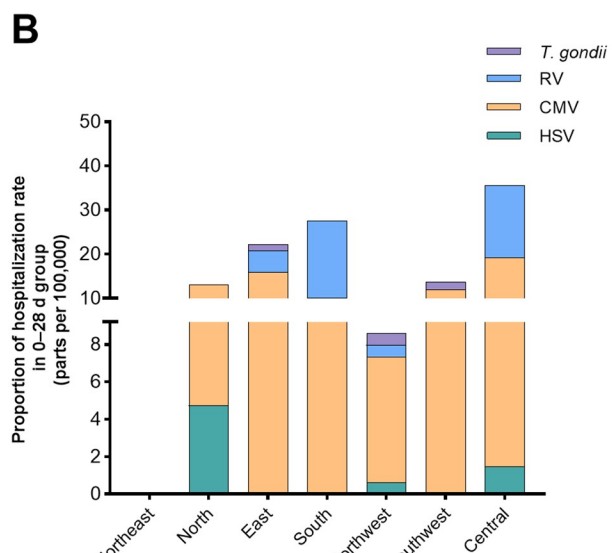

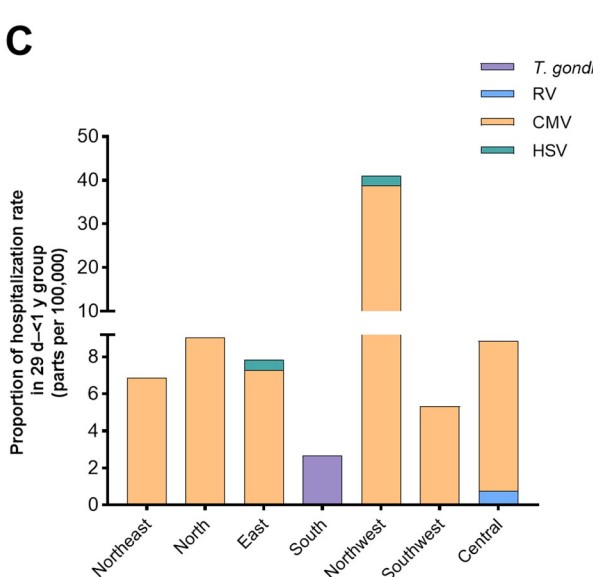

**Fig 1. Proportion of patients hospitalized for congenital TORCH infections.**

China, South China, Northwest China, and Southwest China were 0.13‰ (11/84,291), 0.21‰ (30/144,097), 0.28‰ (11/39,757), 0.09‰ (14/163,123), and 0.14‰ (8/58,083), respectively (Fig 1B). In the cases of infants, patients from the Northwest China had the highest proportion of hospitalization (0.41‰, 35/85275), and most of them were hospitalized with congenital CMV infection. The proportion of hospitalization of infant with congenital TORCH infection in the South China was the lowest (0.03‰, 1/37,510, Fig 1C). The hospitalization proportions of infant patients from Northeast China, North China, East China, Southwest China, and Central China were 0.07‰ (1/14,572), 0.09‰ (7/77,412), 0.08‰ (14/178,740), 0.05‰ (3/56,284), and 0.10‰ (14/135,451), respectively (Fig 1C).

## Congenital toxoplasmosis

Five patients diagnosed with congenital toxoplasmosis included in FRCPD information acquisition and management were reported in this study. The male to female ratio was 1.5:1, 80% of

**Table 2. The general sociodemographic characteristics and disease burden of pediatric patients with *T. gondii* infection during hospitalization from December 2015 to December 2020.**

| No. | Sex | Age (d) | Province (Region) | Ethnicity | Residence | Admission time | Clinical manifestation | LOS (d) | Expense (USD) |
|---|---|---|---|---|---|---|---|---|---|
| 1 | Female | 3 | Yunnan (Southwest China) | Non-Han | Urban | Apr, 2016 | Pneumonia | 5 | 834.31 |
| 2 | Female | 9 | Jiangxi (East China) | Han | Rural | Jan, 2018 | Toxoplasma meningoencephalitis | 22 | 3,446.41 |
| 3 | Male | 37 | Guangdong (South China) | Han | Urban | Jun, 2019 | Unspecified symptoms | 4 | 440.70 |
| 4 | Male | 5 | Hunan (Central China) | Han | Rural | Dec, 2020 | Jaundice | 19 | 2,162.33 |
| 5 | Male | 6 | Jiangxi (East China) | Han | Urban | Aug, 2020 | Toxoplasma meningoencephalitis | 24 | 3,600.19 |

*T. gondii*: *Toxoplasma gondii*

LOS: length of stay

IQR: inter quartile range

USD: US dollar

patients were neonates (0–28 d group), two patients resided were from East China (Jiangxi), and the other three resided in Southwest China (Yunnan), South China (Guangdong), and Central China (Hunan), respectively. The median LOS was 19 days (IQR 4.5–23 days), which was the longest LOS among the hospitalized patients. The median expense of these children was 2,162.33 USD (IQR 637.51–3,523.25 USD). The main clinical manifestation of two patients was *Toxoplasma* meningoencephalitis, and one of them was accompanied by chorioretinitis (Table 2).

## CRS

Twenty-seven patients diagnosed with CRS were reported, accounting for 15.61% of the hospitalizations with congenital TORCH infection. The ratio of males to females was 0.93:1, and the age distribution was mainly in the 0–28 d group (96.30%). The highest hospitalization proportion of neonate patients with CRS was observed in South China, accounting for 0.18‰ (7/39,757) of the total hospitalized neonate cases, followed by Central China (0.16‰, 11/67,524), East China (0.04‰, 7/144,097), and Northwest China (0.006‰, 1/163,123). No hospitalized patients with this diagnosis were reported in Northeast, North and Southwest China (Fig 1B). Only one infant was hospitalized with CRS in the Central China in the 29 d–1 y group (0.007‰, 1/135,451, Fig 1B). The number of hospitalizations in 2019 (especially from September to December) was the highest, the monthly hospitalization rate was 2.5 (admissions per 100,000) in both September (3/10,020) and November (3/10690, S1 Fig). The LOS was 9 d (IQR 5–15 d). Hospitalized children with CRS had the highest expense of 2,585.45 USD (IQR 1,524.88–4,010.98 USD) (Table 1).

The clinical symptom profiles of patients with CRS (including cardiovascular malformation, eye disorder, hearing damage, pneumonia, and neonatal thrombocytopenic purpura) were analyzed by the age category. Among the clinical manifestations of patients with CRS in the 0–28 d group, cardiovascular malformation accounted for the largest proportion (21 cases, 80.77%). One patient was included in the 29 d–<1 y group, and his main clinical manifestations were cardiovascular malformation, eye disorder, and hearing damage (Table 3). Fig 2 showed that CRS was often combined with a variety of clinical manifestations, among which cardiovascular malformation combined with eye disorder was more common (36.00%). According to the statistical analysis, the LOS and expenses were correspondingly increased for the children who were complicated with multiple organ injuries compared with patients with single organ injuries.

**Table 3. The general sociodemographic characteristics of pediatric patients with CRS categorized by clinical manifestations during hospitalization from December 2015 to December 2020.**

|  | Number | Clinical manifestation | | | | | |
|---|---|---|---|---|---|---|---|
|  |  | Cardiovascular malformations | Eye disorder | Hearing damage | Pneumonia | Neonatal thrombocytopenic purpura | Unspecified |
| **Sex (n, %)** |  |  |  |  |  |  |  |
| Male | 13 | 12 (92.31) | 2 (15.38) | 2 (15.38) | 2 (15.38) | 1 (7.69) | 1 (7.69) |
| Female | 14 | 10 (71.43) | 5 (35.71) | 5 (35.71) | 3 (21.43) | 3 (21.43) | 1 (7.14) |
| **Age (n, %)** |  |  |  |  |  |  |  |
| 0–28 d | 26 | 21 (80.77) | 6 (23.08) | 6 (23.08) | 5 (19.23) | 4 (15.38) | 2 (7.69) |
| 29 d–<1 y | 1 | 1 (100.00) | 1 (100.00) | 1 (100.00) | 0 (0.00) | 0 (0.00) | 0 (0.00) |
| **Ethnicity (n, %)** |  |  |  |  |  |  |  |
| Han | 25 | 20 (80.00) | 6 (24.00) | 7 (28.00) | 5 (20.00) | 4 (16.00) | 2 (8.00) |
| Non-Han | 2 | 2 (100.00) | 1 (50.00) | 0 (0.00) | 0 (0.00) | 0 (0.00) | 0 (0.00) |
| **Residence (n, %)** |  |  |  |  |  |  |  |
| Urban | 10 | 8 (80.00) | 1 (10.00) | 2 (20.00) | 2 (20.00) | 1 (10.00) | 1 (10.00) |
| Rural | 17 | 14 (82.35) | 6 (35.29) | 5 (29.41) | 3 (17.65) | 3 (17.65) | 1 (5.88) |

CRS: congenital rubella syndrome

LOS: length of stay

IQR: inter quartile range

USD: US dollar

## Congenital CMV infection

The proportion of children hospitalized with congenital CMV infection was the highest among patients with the four congenital infectious diseases. The male to female ratio was 1.28:1. Neonates accounted for 48.48%, while infant accounted for 51.52% (Table 1). In the cases of neonates, patients with congenital CMV infection have the highest proportion of hospitalized patients in Central China (0.18‰, 12/67,524, Fig 1B). However, in the infant group, the highest rate of hospitalization due to congenital CMV infection was in Northwest China (0.39‰, 33/85,275, Fig 1C). The seasonality could not be assessed because monthly case numbers for congenital CMV infection were rare (S2 Fig).

Our data suggested that the largest number of hospitalized children with CMV infection developed liver damage (36 cases, 27.27%), followed by lung damage (28 cases, 21.21%), blood disorders (11 cases, 8.33%), hearing damage (8 cases, 6.06%), nervous system damage (5 cases, 3.79%), myocardial damage (3 cases, 2.27%), eye damage (1 case, 0.76%), and multiple organ

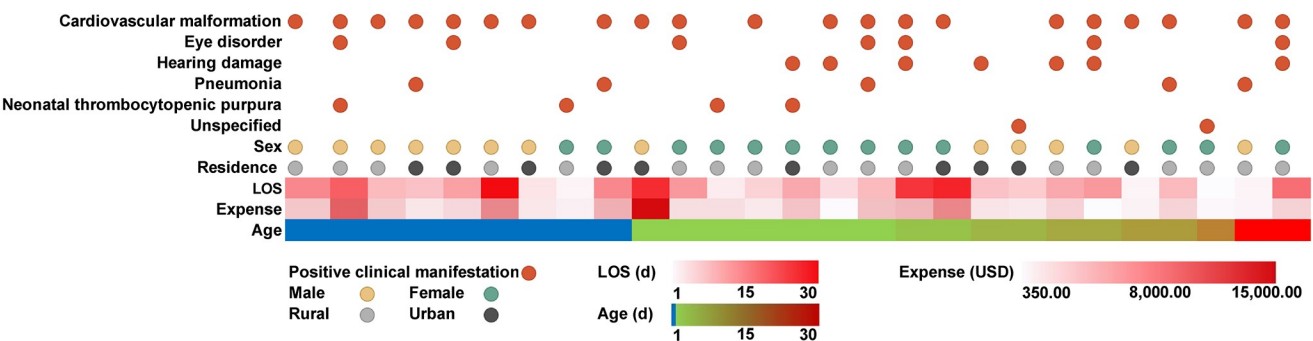

**Fig 2. LOS and expenses of hospitalization due to CRS.**

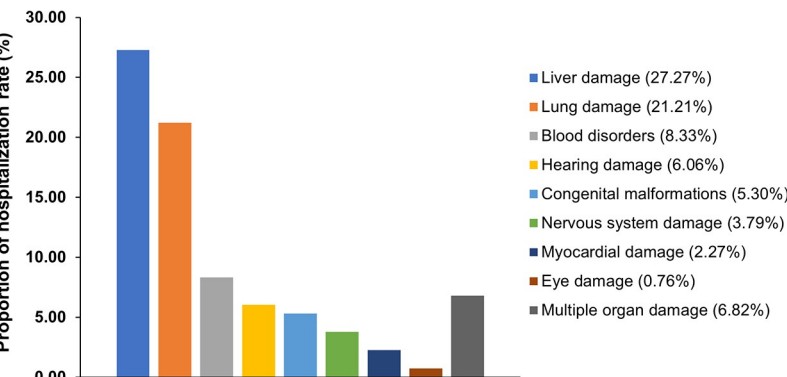

**Fig 3. Proportion of patients with damage to different organs who were hospitalized with congenital CMV infection.**

damage (9 cases, 6.82%). We observed that the LOS and expenses of patients with multiple organ damage were higher than those of patients with single organ injury. The highest expenses appeared in the pneumonia complicated with jaundice and liver damage group (7,205.937 USD) (Fig 3).

## Congenital HSV infection

Nine patients with congenital HSV infection were reported in this study, accounting for 5.20% of the total hospitalized cases with congenital TORCH infection (Table 1). The ratio of males to females was 1.25:1. The age distribution was mainly in the 0–28 d group (66.67%), and the median age was 7 d (IQR 3–29 d). The neonatal cases mainly occurred in North China (0.04‰, 4/84,291, Fig 1B). The major clinical manifestations of these 9 patients are summarized in Table 4. Five patients were mainly characterized by unspecified symptoms, three

**Table 4. The general sociodemographic characteristics and disease burden of pediatric patients with congenital HSV infection during hospitalization from December 2015 to December 2020.**

| No. | Sex | Age (d) | Province (Region) | Ethnicity | Residence | Admission time | Clinical Complication | LOS (d) | Expense (USD) |
|---|---|---|---|---|---|---|---|---|---|
| 1 | Male | 31 | Shaanxi (Northwest China) | Han | Rural | Jun, 2016 | Unspecified | 2 | 287.35 |
| 2 | Male | 3 | Shaanxi (Northwest China) | Han | Rural | Jul, 2016 | Sepsis | 7 | 935.03 |
| 3 | Male | 1 | Inner Mongolia (North China) | Non-Han | Urban | Jul, 2017 | Unspecified | 3 | 724.23 |
| 4 | Female | 31 | Shaanxi (Northwest China) | Han | Urban | Aug, 2017 | Sepsis | 21 | 4197.91 |
| 5 | Female | 29 | Shandong (East China) | Han | Rural | Jun, 2018 | Unspecified | 5 | 724.48 |
| 6 | Male | 16 | Shanxi (North China) | Han | Urban | Oct, 2018 | Jaundice | 29 | 7,127.02 |
| 7 | Female | 1 | Shanxi (North China) | Han | Rural | Jan, 2020 | Unspecified | 6 | 1,319.19 |
| 8 | Male | 7 | Shanxi (North China) | Han | Rural | Jan, 2020 | Unspecified | 1 | 246.72 |
| 9 | Female | 5 | Hubei (Central China) | Han | Rural | Oct, 2020 | Sepsis | 16 | 3,767.50 |
| Median (IQR) | - | 7 (3–29) | - | - | - | - | - | 6 (2.5–18.5) | 935.03 (505.80–3,982.70) |

HSV: Herpes Simplex Virus

LOS: length of stay

IQR: inter quartile range

USD: US dollar

**Table 5. The general sociodemographic characteristics and disease burden of pediatric patients with a mixed infection during hospitalization from December 2015 to December 2020.**

| No. | Sex | Age (days) | Infection | Province (Region) | Ethnicity | Residence | Admission time | Clinical Complication | LOS (d) | Expense (USD) |
|---|---|---|---|---|---|---|---|---|---|---|
| 1 | Female | 3 | RV and HSV | Hunan (Central China) | Han | Rural | May, 2020 | Cardiovascular malformation | 25 | 4,561.07 |
| 2 | Male | 1 | RV and HSV | Guangdong (South China) | Han | Rural | Oct, 2019 | Sepsis | 30 | 7,282.05 |

LOS: length of stay

IQR: inter quartile range

USD: US dollar

patients presented with sepsis, and the remaining patient showed jaundice. The median of LOS and expenses were 6 days (IQR 2.5–18.5 d) and 935.03 USD (IQR 505.80–3,982.70 USD).

## Mixed infection

The presented FSMRs data showed that two patients, diagnosed with CRS, had severe symptoms due to mixed infection with RV and HSV. One of them resided in Central China (3 d, female, Han ethnicity, Hunan, 2020), and another resided in South China (1 d, male, Han ethnicity, Guangdong, 2019). The main complication of the female patient was cardiovascular malformation, and that of the male patient was sepsis. Their LOSs (25 d and 30 d, respectively) and expenses (4,561.07 USD and 7,282.05 USD, respectively) were higher than those of patients with CRS alone (Table 5).

## Discussion

Studies have reported that TORCH infections are currently important risk factors for adverse pregnancy outcomes in China, particularly congenital malformations [22]. This study intuitively describes the epidemiological characteristics and the disease burden of hospitalized children with congenital TORCH infections.

A study of TORCH serological screening in Chinese neonates showed that the seroprevalence of TORCH-IgM was approximately 0.67%. Among these patients, the seroprevalence of CMV was the highest [11, 23]. Our data showed that the hospitalization rate due to congenital TORCH infections was extremely low (17.56 per 100,000 hospitalized neonates). Wang *et al.* reported that the seroprevalence of *T. gondii* among women of childbearing age was 0.35% for IgM and 4.35% for IgG in Xi'an, China [24], which was lower than that in other countries (50–80% in Brazil, 44% in France, and 9.1% in the USA) [25]. Our data showed that only five cases of congenital toxoplasmosis were recorded from 2015 to 2020, indicating that the infection rate of congenital toxoplasmosis in China is indeed at a low level, which may benefit from the increased awareness of protection against toxoplasma infection among women of childbearing age (for example avoiding contact with pets and eating raw meat). In contrast to the low hospitalization rate, the disease burden of congenital toxoplasmosis was considerable. The LOS for congenital toxoplasmosis was the highest, and the hospitalization expense was also in the second position among congenital TORCH infectious diseases, indicating that congenital toxoplasmosis, although uncommon, may cause severe clinical symptoms and a heavy disease burden. Torgerson *et al.* reported that the global burden of congenital toxoplasmosis was estimated to be 9.6 disability-adjusted life years (DALYs) per 1,000 live births [3]. Therefore, more attention should be given to strengthening prepregnancy toxoplasmosis detection and

educating people on toxoplasmosis prevention knowledge to reduce the incidence of congenital toxoplasmosis.

RV remains an important pathogen worldwide, with an estimated 100,000 cases of CRS per year [26]. Our results showed that the proportion of CRS cases was 0.7 per 100,000 hospitalized neonates, which was higher than that in England (0.18 per 100,000) [27]. The implementation of immunization programs with two doses of the measles–mumps–rubella (MMR) vaccine in China occurred later than that in the UK and might be responsible for the result. Nevertheless, the number of CRS cases has decreased significantly since the universal vaccination began in China in 2008 [7]. A large RV outbreak that occurred in China during 2019, which primarily involved primarily adolescents and reproductive-aged adults [7]. A total of 32,539 cases (2.33 per 100,000) were reported, and most of them occurred from March to June. Among them, 98.41% of patients with rubella over 20 years old had no or unknown immunization history with rubella and the rubella-containing vaccines (RCV), and the vast majority of women who are currently of childbearing age have not been vaccinated against RCV [28]. Our statistics showed that 59.26% of neonatal patients with CRS were hospitalized in 2019. Markedly, most cases occurred from September to December, indicating that their mother might have been infected with RV in early pregnancy, resulting in CRS. A nationwide rubella antibody serological survey of approximately 800,000 women of childbearing age showed that the positive rate of rubella antibodies among Chinese women of childbearing age between 20 and 29 y was only 59%, which did not reach 85% of the population to form an effective immune barrier [29]. Achieving complete anti-rubella vaccination coverage for uncovered adolescent girls and women of childbearing age along with a universal immunization program consisting of a MMR vaccine is paramount. The disease burden of CRS may be very substantial due to the severe clinical symptoms caused by CRS (cardiovascular malformation, cataracts, and deafness) [30]. In the present study, 80% of hospitalized children with CRS had cardiovascular malformations, indicating that RV is a risk factor for neonatal teratogenicity, and that high surgical treatment costs will be incurred. The economic burden of CRS was heavy, and infants are recommended be vaccinated against RV in a timely manner [31]. Strengthening CRS surveillance is conducive to the rapid identification and control of rubella outbreaks, preventing the secondary transmission of rubella due to CRS cases, and has important value for prevention and control significance.

Congenital CMV infection is the most common congenital infection worldwide. The prevalence of congenital CMV infections is approximately 0.2% to 2% (average 0.65%) in developed countries and 6–14% in developing countries [32]. According to a previous study, the CMV seroprevalence was 98.11% and the congenital CMV prevalence was 1.32% in China [33]. Our statistical analyses showed that children with congenital CMV infection had the highest hospitalization rate among those with TORCH infection. Furthermore, the proportion of hospitalizations for congenital CMV infections in the 29 d–<1 y group was similar to that in the 0–28 d group, possibly because most neonates with congenital CMV infection detected at birth were asymptomatic and were not admitted until obvious clinical manifestations occurred, such as liver damage. In addition, the proportion of children with congenital malformations was 4.76%, suggesting that the teratogenic effect of congenital CMV infection must not be ignored, and the prevention of congenital CMV infection during pregnancy is still an important issue that should be considered. Routine screening for neonatal congenital CMV infection and proactive preventive and therapeutic measures might help reduce the disease burden caused by congenital CMV infection. Congenital CMV infection can induce hearing loss; however, only 5.54% of children present hearing damage. We speculated that the reason may be the lack of outpatient information and follow-up information. Only FSMRs of hospitalized children were collected in this study, and the effect of the congenital CMV infection probability might be underestimated.

Studies have reported that the incidence of neonatal HSV infection is relatively low, occurring in 1 in 3,000 to 20,000 livebirths [34][13]. The incidence in China is unknown due to a lack of reported data on neonatal HSV infection. A study indicated that between 2008 and 2010 in Hong Kong, China, only one case of the neonate was infected with HSV [22]. In our study, the proportion of hospitalizations for congenital HSV infections was 1.05/100,000 neonatal patients, indicating that the incidence of congenital HSV infection is comparatively low in China. Neonatal disseminated HSV infection may cause viral sepsis, liver failure, etc., with the worst prognosis [35]. According to our findings, the disease burden (including LOS and expenses) increases significantly when patients develop complications. Early diagnosis and timely treatment of neonatal HSV infection are very important to improve national population quality. We also identified mixed infections with HSV and RV; the explanation might be that there may be a certain mutual activation may exist between various pathogens causing TORCH infections [36], and its mechanism requires further exploration.

Some limitations existed in this study. First, we only collected FSMRs for hospitalized children with congenital TORCH infection and lacked a considerable amount of medical information on children attending outpatient clinics, as well as follow-up information for these children. Second, this study only statistically analyzed the hospitalization due to four major pathogens causing TORCH and did not include other pathogens, such as parvovirus B19, herpes zoster virus, and syphilis (it was monitored by an independent national surveillance network). Therefore, the effect of congenital TORCH infection may be underestimated. Third, the FUTURE database does not contain a detailed course of disease, laboratory examination results, and treatment records of patients, which may lead to the simplified and insufficiently comprehensive of the data. Fourth, the database was unable to collect information on all patients with mild symptoms of TORCH infection and the loss of infants who died secondary to their infection or fetal stillbirth, and the epidemiological burden of congenital TORCH infections is underestimated. Finally, because of the number of hospitalized patients with some diseases was relatively rare, the calculated disease burden may differ from the actual situation, and requires an expanded version of the FUTURE database to reinforce the conclusions obtained in this study.

## Conclusion

We combed the FUTURE database of 27 children's hospitals in China and described the general epidemiological characteristics and the disease burden of hospitalized children infected with congenital TORCH (*T. gondii*, RV, CMV, and HSV) infection. The hospitalization rate due to congenital TORCH infections was extremely low (17.56 per 100,000 hospitalized neonates). However, they still imposed a significant medical burden on clinical and public health in China. These data illustrate the importance of improving screening for congenital TORCH infections in the early diagnosis and treatment of neonatal patients. Increased awareness of the epidemiological characteristics and the disease burden of congenital TORCH infections is conducive to public health education (promoting antenatal hygiene behaviors and immunization), formulating targeted strategies (utility of neonatal screening), and research (through vaccine development).

## Supporting information

**S1 Fig. Hospitalization rate and number of neonates with CRS (n = 26), December 2015–December 2020.** M (n): number of monthly hospitalizations, Y (n): number of yearly hospitalizations.
(TIF)

**S2 Fig. Hospitalization rate and number of congenital CMV infections (n = 132), December 2015–December 2020.** M (n): number of monthly hospitalizations, Y (n): number of yearly hospitalizations.
(TIF)

## Acknowledgments

We are grateful to investigators from members of the Futang Research Center of Pediatric Development (FRCPD).

## Author Contributions

**Conceptualization:** Linlin Zhang, Ran Wang, Zhengde Xie.

**Data curation:** Guoshuang Feng, Yueping Zeng.

**Formal analysis:** Xinyu Wang.

**Funding acquisition:** Ran Wang, Zhengde Xie.

**Investigation:** Linlin Zhang, Xinyu Wang, Mengjia Liu.

**Methodology:** Xinyu Wang, Mengjia Liu.

**Project administration:** Ran Wang.

**Resources:** Xinyu Wang, Zhengde Xie.

**Supervision:** Ran Wang, Zhengde Xie.

**Validation:** Xinyu Wang, Guoshuang Feng, Yueping Zeng, Ran Wang, Zhengde Xie.

**Visualization:** Linlin Zhang, Ran Wang.

**Writing – original draft:** Linlin Zhang, Ran Wang.

**Writing – review & editing:** Xinyu Wang, Ran Wang, Zhengde Xie.

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
