## [Decision Letter · Decision Letter 0]

15 Aug 2022

Dear Dr. Wang,

Thank you very much for submitting your manuscript "The epidemiology and disease burden of congenital TORCH infections among hospitalized children in China: A national cross-sectional study" for consideration at PLOS Neglected Tropical Diseases. As with all papers reviewed by the journal, your manuscript was reviewed by members of the editorial board and by several independent reviewers. In light of the reviews (below this email), we would like to invite the resubmission of a significantly-revised version that takes into account the reviewers' comments. 

The authors have described the disease burden of congenital TORCH infections and described their clinical features from a nationwide database. There are considerable methodological constraints which require clarity. The authors need to clarify how the diagnosis of a congenital TORCH infection was made. As has been pointed out by the reviewers as well, the basis of diagnosis of TORCH infection especially that of congenital CMV infection need to be described. Congenital CMV is the commonest congenital infection described in the study and over 50% of the children with congenital CMV infection are beyond the neonatal period. A serological or PCR based diagnosis made beyond 3 weeks cannot differentiate congenital from acquired CMV infection. There are other issues that have been raised by the reviewers and the manuscript requires considerable revision before it can be accepted.

We cannot make any decision about publication until we have seen the revised manuscript and your response to the reviewers' comments. Your revised manuscript is also likely to be sent to reviewers for further evaluation.

Sincerely,

Winsley Rose

Academic Editor

Marcelo Ferreira

Section Editor

The authors have described the disease burden of congenital TORCH infections and described their clinical features from a nationwide database. There are considerable methodological constraints which require clarity. The authors need to clarify how the diagnosis of a congenital TORCH infection was made. As has been pointed out by the reviewers as well, the basis of diagnosis of TORCH infection especially that of congenital CMV infection need to be described. Congenital CMV is the commonest congenital infection described in the study and over 50% of the children with congenital CMV infection are beyond the neonatal period. A serological or PCR based diagnosis made beyond 3 weeks cannot differentiate congenital from acquired CMV infection. There are other issues that have been raised by the reviewers and the manuscript requires considerable revision before it can be accepted.

Reviewer's Responses to Questions

**Key Review Criteria Required for Acceptance?**

**Methods**

-Are the objectives of the study clearly articulated with a clear testable hypothesis stated?

-Is the study design appropriate to address the stated objectives?

-Is the population clearly described and appropriate for the hypothesis being tested?

-Is the sample size sufficient to ensure adequate power to address the hypothesis being tested?

-Were correct statistical analysis used to support conclusions?

-Are there concerns about ethical or regulatory requirements being met?

Reviewer #1: The authors describe the epidemiology and the disease burden of congenital TORCH infections (Toxoplasmosis, CMV, Rubella, HSV) among hospitalized infants in China. The data has been derived from the FUTURE database which collects data from 27 centres from various regions of China. The authors have stated that the data was on all infants admitted to these hospitals between 2015 and 2020.

Since they have limited their objectives to number of admissions with the TORCH infections, length of stay and treatment cost, this seems reasonable.

However, to make the subsequent data more meaningful, some more background data needs to be given:

1. How were the diagnosis of each of these infections established? Only serological tests or were molecular and/or cultures used? 

2. Were all the diagnoses made in the newborn period? Especially for CMV, if diagnosis was made after 3 weeks , it is likely to be a postnatal infection. This distinction, to a lesser extent, has to be made for rubella and HSV also. So, the authors have to be categorical that the diagnosis of a CONGENITAL infection was established even though the child may have been admitted at a later age. 

3. It is very unclear from the write up if children below 1 year were included or all children ≤ 18 years were included in the analysis. The inclusion criteria says ≤18 years, but the tables all use 29 days- 1 year. The year by year analysis and the month-wise admissions would have little meaning if ALL children ≤18 years were included. 

4. The exclusion criteria excludes children where data regarding age, gender, burden of disease, diagnosis were not complete. While it makes perfect sense to exclude those with incomplete diagnosis, it would be interesting to see the numbers of those excluded on the basis of lack of other criteria especially if the numbers are large.

5. Since the authors have chosen to determine the ethnicity of the patients with infections, it is worthwhile giving the ethnic diversity (Han vs Non-Han) of China as a whole or within the population analyzed. If this is almost the same as the proportion among the infected children, then there is no ethnic variation.

Reviewer #2: This study is a descriptive study, but the field is poorly studied from an epidemiologic standpoint. The true burden of TORCH infections is under characterized and this study adds value into this field. The statistical methods are relatively simple as a descriptive study and appropriate. The data seem to be abstracted from a discharge face sheet which is not clear to me what this contains- I suspect it is akin to a US medical system discharge summary, but the data abstraction should be better described.

**Results**

-Does the analysis presented match the analysis plan?

-Are the results clearly and completely presented?

-Are the figures (Tables, Images) of sufficient quality for clarity?

Reviewer #1: To make better sense of the vast data collected, I would like more data given. 

1. How many hospitalized children were there in each age group over the 5 years? If we have the number of births/ newborn in the 27 hospitals and the number of neonatal diagnosis of the TORCH infections, this would give a better indicator of the burden of the infections. 

2. There is a wide disparity in the distribution of hospitals between the regions- 8 in East China and 2 each in South and South-West China. Hence it does not make sense to give the percentage of each infection as a proportion of the total infections in each region. If the population studied in each region is taken as the denominator, then we would be able to glean if an infection is in fact more prevalent in a particular region. Hence, the way this is given in the text and the tables now should be removed or modified.

3. As previously mentioned, the season of admission may be valid if only neonatal infections are included. Otherwise, giving this data for all children between 0-12 months gives no meaningful information. A baby born in one particular month could be admitted with cardiac or liver cell failure any time over the next year! If this data is to be included, it should changed to include only newborn. That would give some idea of the seasonality of the infection in the mother.

Tables and figures:

1. The tables are exhaustive and can be made more brief to give only the relevant information. For example, the clinical manifestations in various organs in CRS (Table 3 ) in each region is redundant. As previously mentioned, ethnicity would have relevance if the background ethnic make-up is known. 

2. The supplementary figures S1 and S2 are too unclear. No details could be made out.

Reviewer #2: The analysis and the analysis plan matches. 

There are some significant limitations in the data- Parvovirus should be excluded as parvovirus has implications in vertical transmission antenatally and can cause hydrops, stillbirth and severe anemia. Post natal admission for parvovirus would be atypical and the authors should exclude it. 

It is difficult to believe that there was no vertical transmission of HSV. This is a relatively common TORCH pathogen and certainly exists at higher rates than congential toxo.

**Conclusions**

-Are the conclusions supported by the data presented?

-Are the limitations of analysis clearly described?

-Do the authors discuss how these data can be helpful to advance our understanding of the topic under study?

-Is public health relevance addressed?

Reviewer #1: If necessary changes as suggested could be made, the conclusions would be more clear. The authors have clearly identified the limitations of the analysis and have described them. I would like some more detailing on how they expect this data to have public relevance in China.

Reviewer #2: In the conclusions, the authors perform a comparative analysis of different TORCH infections. This is likely inappropriate due to the wide variety in clinical sequelae, clinical presentation and antenatal route of infection. 

In the section on CRS, additional information on maternal immunization history and rubella susceptibility in the setting of the described outbreak is warranted. Vaccination is a well described strategy to eliminate CRS and should be commented on. 

The authors should describe the epidemiology of postnatal serology and the difference in serologic titers (particularly positive IgG) and congenital TORCH infection diagnosis. Both background on transplacental passage and stability of IgG should be described as well as a description on the utility of serology for postnatal screening as this is elucidated in several areas. 

Several limitations of this study should be discussed in the conclusions including the inability to detect mildly symptomatic infections for all TORCH pathogens as well as the loss of infants who died secondary to their infection or fetal stillbirth. While this dataset is extremely important and adds to the data, it underrepresents the epidemiologic burden of TORCH and this should be explicitly stated

**Editorial and Data Presentation Modifications?**

Reviewer #1: More data needs to be presented before the conclusions can be arrived in this study. The language used needs a fair bit of editing before publication.

Reviewer #2: The introduction is very short and largely written in a matter of fact matter without a lead into the subject material and study desigen. I would recommend additional background and information should be included to introduce and link the study.

**Summary and General Comments**

Reviewer #1: The authors have presented data on congenital TORCH infections among hospitalized children in China over a nearly 6 year period from China. It should give a good idea about the burden of the problem (health and economic) and the prevalence of these infections in China to direct public health measures. I think this would be good data.

 However, more background data needs to be given and the analysis needs to be modified to make the results and conclusions more valid.

Reviewer #2: The manuscript by Dr. Zhang and colleagues about the epidemiology and burden of TORCH infections is a novel study that substantially adds to information in the field on this topic. The study design is a retrospective collection of data from children age 0-1 with a diagnosis code of a congenital infection. They identified 173 patients from a wide and diverse geographic region in China from their database. They then go to describe the distribution attributable to different pathogens and some information on the sequelae of infection. 

This manuscript would benefit from additional detail regarding serologic screening, background on the diagnosis of vertical transmission versus postnatal transmission (well described in CMV for example) as well as an additional figure with the geographic region of each infection (as this is an epidemiologic study and local geopolitical and cultural risks may contribute to disease prevalence).

PLOS authors have the option to publish the peer review history of their article (what does this mean?). If published, this will include your full peer review and any attached files.

Reviewer #1: No

Reviewer #2: Yes: Christina Megli MD/PhD
---

## [Decision Letter · Decision Letter 1]

3 Oct 2022

Dear Dr. Wang,

We are pleased to inform you that your manuscript 'The epidemiology and disease burden of congenital TORCH infections among hospitalized children in China: A national cross-sectional study' has been provisionally accepted for publication in PLOS Neglected Tropical Diseases.

Best regards,

Winsley Rose

Academic Editor

Marcelo Ferreira

Section Editor

The revised version is accepted for publication.

Reviewer's Responses to Questions

**Key Review Criteria Required for Acceptance?**

**Methods**

-Are the objectives of the study clearly articulated with a clear testable hypothesis stated?

-Is the study design appropriate to address the stated objectives?

-Is the population clearly described and appropriate for the hypothesis being tested?

-Is the sample size sufficient to ensure adequate power to address the hypothesis being tested?

-Were correct statistical analysis used to support conclusions?

-Are there concerns about ethical or regulatory requirements being met?

Reviewer #1: The objectives, study design and statistical analysis are appropriate.

There are no ethical concerns.

Reviewer #2: Yes- the study is a novel observational study. They clearly describe the methodology.

**Results**

-Does the analysis presented match the analysis plan?

-Are the results clearly and completely presented?

-Are the figures (Tables, Images) of sufficient quality for clarity?

Reviewer #1: Overall, the results in the edited version are well presented. I would suggest some changes in the manuscript:

1. In the results section (Lines 217-225), the authors could change the terminology to "the proportion of hospitalizations for congenital TORCH infections per 100,000 patients was 17.56 (100/569,680) in the 0–28 d group (neonate)" to "the proportion of hospitalizations for congenital TORCH infections was 17.56/ 100, 000 newborn (or alternatively as "per 100,000 live births" if most newborn were inborn in the hospitals) (100/569,680)." For all the individual TORCH infections too, it makes more sense to give the numbers as "per 100,000 newborn or live births". For the over 1 month group, they can quote as per 100,000 admissions. Please add the denominators to each individual result.

2. A query about figure 1: There is a gross discrepancy between the admissions in the neonatal period and later admissions in South and Northwest regions. It is difficult to imagine that none of the newborn with CRS or CMV in South China needed readmission. The exact opposite in the Northwest. Though the number of newborn diagnosed with TORCH is less, it has the highest number of childhood admissions in the country. Is there a difference in testing procedures/ admission criteria in the Northwest? Can the authors recheck these results? And, if possible, give some sort of explanation for these?

3. In an article which gives nationwide data on TORCH infections, it makes little sense to give the individual patient characteristics of Toxoplasmosis, HSV and mixed (Tables 2, 4 and 5). They do not add much information. The data is summarized in the text. These seem to be redundant and may be removed or put as supplementary material.

Reviewer #2: The figures are comprehensive and excellent summary of the data. The results are clearly described and presented. This is a substantial improvement from previous version.

**Conclusions**

-Are the conclusions supported by the data presented?

-Are the limitations of analysis clearly described?

-Do the authors discuss how these data can be helpful to advance our understanding of the topic under study?

-Is public health relevance addressed?

Reviewer #1: No issues

Reviewer #2: The data are supported by the conclusions and the limitations and analysis are clearly presented. The revisions have provided substantial clarity and language improvements.

**Editorial and Data Presentation Modifications?**

Reviewer #1: I think the article needs minor revisions as described above.

Reviewer #2: (No Response)

**Summary and General Comments**

Reviewer #1: The authors have presented data on congenital TORCH infections among hospitalized children in China over a nearly 6 year period. It should give a good idea about the burden of the problem (health and economic) and the prevalence of these

infections in China to direct public health measures. In the present format, the authors have detailed the issues with each of the TORCH infections well.

Reviewer #2: The authors have thoughtfully and extensively addressed the concerns of the reviewers and have produced a comprehensive manuscript that is interesting and somewhat novel to describe the burden of TORCH infections in China. They have clearly described epidemiologic, cost and burden of disease within the limitations of their database. The result is a significant and novel publication that will be of interest.

PLOS authors have the option to publish the peer review history of their article (what does this mean?). If published, this will include your full peer review and any attached files.

Reviewer #1: **Yes: **Sridhar Santhanam

Reviewer #2: **Yes: **Christina Megli

---

## [Editor Report · Acceptance letter]

8 Oct 2022

Dear Dr. Wang,

We are delighted to inform you that your manuscript, "The epidemiology and disease burden of congenital TORCH infections among hospitalized children in China: A national cross-sectional study," has been formally accepted for publication in PLOS Neglected Tropical Diseases.

Best regards,

Shaden Kamhawi

co-Editor-in-Chief

Paul Brindley

co-Editor-in-Chief
